# Cucurbitacin B-, E-, and I-Induced Browning of White Adipocytes Is Promoted by the Inhibition of Phospholipase D2

**DOI:** 10.3390/ijms232315362

**Published:** 2022-12-06

**Authors:** Sun Young Park, Hye Mi Kang, Jin-Woo Oh, Young-Whan Choi, Geuntae Park

**Affiliations:** 1Bio-IT Fusion Technology Research Institute, Pusan National University, Busan 46241, Republic of Korea; 2Department of Horticultural Bioscience, Pusan National University, Myrang 50463, Republic of Korea; 3Department of Nanofusion Technology, Pusan National University, Busan 46241, Republic of Korea

**Keywords:** cucurbitacin, beige adipocyte, UCP1, mitochondria biogenesis, phospholipase D2

## Abstract

The mechanism of white adipose tissue browning is not well understood; however, naturally occurring compounds are known to play a positive role. The effects of cucurbitacins B, E, and I on the browning of mature white adipocytes were investigated. First, the cell viability exhibited by cucurbitacins B, E, and I in pre- and mature adipocytes was verified. Cucurbitacins B, E, and I had no effect on cell viability in pre- and mature adipocytes at concentrations up to 300 nM. To investigate the characteristics of representative beige adipocytes, the formation and morphology of cucurbitacin B, E, and I lipid droplets were verified. The total lipid droplet surface area, maximum Feret diameter, and total Nile red staining intensity of cucurbitacin B-, E-, and I-treated adipocytes were lower than those of mature white adipocytes. Furthermore, treatment of white mature adipocytes with cucurbitacin B, E, and I led to the formation of several small lipid droplets that are readily available for energy expenditure. We evaluated the effect of cucurbitacins B, E, and I on the expression of representative browning markers UCP1, PGC1a, and PRDM16, which participate in the browning of white adipose tissue. Cucurbitacins B, E, and I increased the mRNA and protein expression levels of UCP1, PGC1a, and PRDM16 in a concentration-dependent manner. To promote energy consumption by beige adipocytes, active mitochondrial biogenesis is essential. Next, we investigated the effects of cucurbitacin B, E, and I on mitochondrial biogenesis in mature adipocytes. Mitochondrial mass increased when mature adipocytes were treated with cucurbitacin B, E, and I. The degree of cucurbitacin B-, E- and I-induced transformation of white adipocytes into beige adipocytes was in the order of Cu E > Cu B > Cu I. To verify the effect of phospholipase D2 on the browning of white adipocytes, CAY10594—a PLD2 pharmacological inhibitor, and a knockdown system were used. PLD2 inhibition and knockdown improved the expression levels of UCP1, PGC1a, and PRDM16. In addition, PLD2 inhibition and knockdown in mature white adipocytes promoted mitochondrial biosynthesis. The effect of PLD2 inhibition and knockdown on promoting browning of white adipocytes significantly increased when Cu B, Cu E, and Cu I were co-treated. These data indicate that mature white adipocytes’ beige properties were induced by cucurbitacins B, E, and I. These effects became more potent by the inhibition of PLD2. These findings provide a model for determining anti-obesity agents that induce browning and increase energy expenditure in mature white adipocytes.

## 1. Introduction

Obesity, an immense social burden, is usually caused by increased energy intake through diet and decreased physical activity associated with modern lifestyles. According to the World Health Organization (WHO), there are more than 1 billion obese people worldwide, including 650 million adults, 340 million adolescents, and 39 million children. By 2025, approximately 167 million adults and children worldwide, who are overweight or obese, will suffer from diminished health (WHO, 2022) [1]. Obesity is known to be the cause of 80% of diabetes and 21% of heart disease cases. In addition, in the case of obesity or overweight patients, the incidence rates of osteoarthritis, sleep apnea, and cholelithiasis are higher than those in non-obese people. As such, obesity has become a major public health problem because it has a greater impact on morbidity than on mortality [2,3]. Recently, various methods such as diet therapy, exercise therapy, lifestyle modification, drug therapy, and surgical treatment have been implemented to treat obesity. Many anti-obesity drugs are being developed every year for the treatment of obesity; however, due to various side effects, currently available anti-obesity drugs are limited [4,5]. In addition, most anti-obesity drugs are limited to appetite suppressants, classified as psychotropic drugs and digestive suppressants, with side effects such as diarrhea and constipation. These anti-obesity drugs have been reported to have various side effects such as bronchodilation, addictive substances, and worsening of PMS symptoms [6,7]. Therefore, there is an urgent need to develop a safer compound with an anti-obesity effect equal to or greater than that of existing anti-obesity treatments using natural ingredients.

Adipocytes, which play an important role in energy homeostasis, are closely related to metabolic diseases such as obesity and diabetes [4]. In the study of obesity control mechanisms, strategies for the potential treatment of obesity and related diseases in the browning of white adipose tissue have been selected in recent years. Beige adipocytes are genetically intermediate between white and brown adipocytes but can have energy-burning performance at levels comparable to those of classic brown adipocytes [8,9]. The browning of white adipocytes differs in the structure of organelles, a typical feature of which is the formation of many small lipid droplets. Beige adipocytes due to browning have a relatively large number of mitochondria compared to white adipocytes and are colored brown by the cytochrome pigment present in the mitochondria. Beige adipocytes also undergo the process of optimizing fat breakdown to provide fuel for heat production [8,10]. The expression of UCP-1 in adipocytes with multivesicular lipid droplets, including the ability to generate heat, is termed as beige adipocytes. Beige adipocyte UCP-1 plays an important role in thermogenesis by consuming mitochondrial oxidative energy rather than storing it by releasing it as heat rather than using it to generate ATP. Beige adipocytes have the characteristics of UCP-1 expression and mitochondrial production and express various beige-adipocyte-specific markers such as PRDM16 and PGC1a [7,11,12]. Brown and beige adipocytes contribute to the regulation of energy consumption in the body; thus, promoting the differentiation of white adipocytes into brown and beige adipocytes can help alleviate the side effects of white fat and improve various metabolic diseases.

Cucurbitacin is an oxygenated tetracyclic triterpene found in many pumpkins and plants, including cucumbers, melons, watermelons, and melons of the genus Cucumis. According to the structure, approximately 20 types of cucurbitacin compounds, A to T, have been identified [13,14]. Cucurbitacin compounds, which are representative components of these Cucurbitaceae plants, mainly produce a bitter taste, and through this, they are utilized as a defense mechanism in insects or hostile plants. In addition, cucurbitacin has excellent pharmacological potential, including anti-inflammatory, antioxidant, anticancer, antidiabetic, anticoagulant, and anti-arteriosclerosis; thus, its usefulness is very high [15,16]. Accumulated evidence suggests that CuB possesses a variety of biological activities, including antidiabetic, anti-inflammatory, and anti-tumor activities, as well as the prevention of atherosclerosis and protection of the liver. CuB has also been reported to inhibit the development of leukemia and breast, lung, and liver cancers [17,18]. CuE has demonstrated excellent antipyretic and anti-inflammatory effects, and has been proven to relieve symptoms through antioxidant and anti-inflammatory properties in testosterone-induced prostatic hyperplasia. It has also been reported to attenuate the malignant progression of cancer through various mechanisms, such as inhibition of proliferation and invasion of cancer cells, induction of apoptosis, and immune regulation [15,19,20]. CuI, a JAK2-STAT3 signaling pathway inhibitor, has been demonstrated to have antioxidant, immune, gastrointestinal, and neuroinflammation-inhibitory effects. In addition, it has an antitumor mechanism of action through the inhibition of various signaling pathways in xenograft and cell cycle models [15]. The 3T3-L1 murine pre-adipocyte is a well-established in vitro model for evaluating the potential adipogenesis-inhibitory effects of natural compounds on adipose tissue. 3T3-L1 murine pre-adipocytes were differentiated into mature adipocytes after hormonal stimulation for more than nine days in MDI differentiation medium (dexamethasone, IBMX, insulin, and rosiglitazone) [21,22]. Published research results have revealed that Cu B, Cu E, and Cu I suppress obesity through the inhibition of adipocyte differentiation [23,24]. Based on this, the aims of this study were to investigate the potential effects of Cu B, Cu E, and Cu I on browning of mature adipocytes and the potential mechanism, particularly its underlying molecular mechanisms, in 3T3-L1 murine pre-adipocytes. To the best of our knowledge, this study is the first comparative analysis of the roles of Cu B, Cu E, and Cu I in the browning of mature adipocytes, and on the mechanism of browning of cucurbitacin B, E, and I in white adipocytes. We focused on relevant targets, pathways, and specific relationships between cucurbitacin B, E, and I and phospholipase D2. In addition, various experiments were conducted to validate the results.

## 2. Results

### 2.1. Effect of Cu B, Cu E, and Cu I on Nile Red-O Staining Activity and Cell Viability

Cucurbitacin, a major metabolite of various Cucurbitaceae plants, is a highly oxygenated tetracyclic triterpenoid comprising 19-(10 → 9β)-abeo-10α-lanost-5-ene. It also has a highly unsaturated backbone and various hydroxyl, acetoxy, and ketone groups [13,16]. The chemical and physical properties of cucurbitacins B, E, and I (CuB, CuE, and CuI) are listed in Appendix A. Their chemical structures are shown in Figure 1A. The CCK-8 assay was used to determine the effects of Cu B, Cu E, and Cu I on cell viability of pre- and mature adipocytes. CCK-8, which provides a convenient and powerful method for cell viability analysis, uses a water-soluble tetrazolium salt to quantify the number of living cells and bio-reduction in the presence of electron carriers, and then generates orange formazan dye and analyzes it through a final colorimetric method [25]. In the range of 50–300 nM, there was no toxicity to CuB, CuE, or CuI, and no change in cell viability was observed in proportion to the concentration (Figure 1B). Therefore, in subsequent experiments, the concentrations of CuB, CuE, and CuI were chosen as 50–200 nM. To compare the effects of CuB, CuE, and CuI on adipogenesis inhibition, lipid accumulation was monitored by flow cytometry using a Nile red staining assay. Nile red, 9-diethylamino-5H-benzoαphenoxazin-5-one, is lipophilic—that is, it binds to intracellular neutral lipids and forms a linear correlation between Nile red fluorescence and neutral lipid content observed via confocal microscopy and flow cytometry [26]. When mature adipocytes were treated with 100 and 200 nM CuB, CuE, and CuI, lipid accumulation decreased in a concentration-dependent manner. A comparison of the lipid accumulation inhibitory levels of CuB, CuE, and CuI confirmed that they had an inhibitory effect in the order of CuE > CuB > CuI at 100 nM, and it was confirmed that they had a similarly high inhibitory effect at 200 nM (Figure 1C). The above results reconfirmed that CuB, CuE, and CuI reduced the accumulation of lipids in mature adipocytes [23,24], similar to previously published results.

### 2.2. Effect of Cu B, Cu E, and Cu I on Formation of Smaller Lipid Droplets

Treatment of pre-adipocytes with MDI leads to white adipogenesis and the presence of single lipid droplets in almost all regions of nucleated adipocytes. However, when the change to beige or brown adipocytes is achieved, several small lipid droplets are formed around the nucleus. In other words, during lipolysis, large lipid droplets break down and several small lipid droplets are generated [11,27]. Confocal microscopy revealed that large lipid droplets were formed in differentiated adipocytes. In contrast, in terms of the dynamics of the lipid droplets, the surface area and size of the lipid droplets decreased in the Cu B, Cu E, and Cu I treatment groups (Figure 2A). This is a unique characteristic of beige adipocytes. In addition, as a result of comparing the total Nile red staining intensity through a flow cytometer, it was confirmed that the Nile red staining intensity of CuB, CuE, and CuI was reduced by more than 3.9 times compared to the control group (Figure 2B). As a result of comparing the total lipid droplets surface area, the control showed a surface area of 243.8 μM^2^, whereas CuB showed 73.9 μM^2^, CuE 67.6 μM^2^, and CuI 82.8 μM^2^ (Figure 3C). In the case of Maximum Feret diameter, the control showed a diameter of 15.3 μM, whereas CuB showed 6.5 μM, CuE 6.2 μM, and CuI 9.5 μM (Figure 3D). By measuring the total Nile red staining intensity through confocal microscopy, it was confirmed that CuB, CuE, and CuI were reduced by more than two times compared to the control group (Figure 3E). These results confirmed the effect of Cu B, Cu E, and Cu I on the formation of lipid droplets characteristic of brown- and beige-related adipocytes.

### 2.3. Effect of Cu B, Cu E, and Cu I on Expression Levels of Brown- and Beige-Fat Specific Marker Genes

The expression of UCP-1, a thermogenic gene, in brown adipocytes is an essential prerequisite for promoting energy consumption in beige adipocytes. Brown and beige adipocytes expressing UCP-1 play an important role in metabolic activity [8,28]; therefore, it is necessary to verify Cu B, Cu E, and Cu I, which increase the number of brown and beige adipocytes. RT-qPCR and Western blotting analyses were performed to measure the increase in the expression of UCP-1, which plays an important role in the change from white adipocytes to brown adipocytes, such as an increase in energy consumption efficiency. The results confirmed that the transcription and translation levels of UCP-1 increased in a concentration-dependent manner after 50–200 nM Cu B, Cu E, and Cu I. In addition, it was confirmed that the expression of UCP-1 increased in the order Cu E > Cu B > Cu I at the same concentration of 200 nM (Figure 3A,B). In addition, PRDM16 and PGC1a have been reported to be transcriptional regulatory factors that regulate the formation of brown and beige adipocytes in mature white adipocytes. PRDM16 induces the differentiation of white adipocytes into brown adipocytes and induces the expression of its partner PGC1a, which subsequently induces an increase in UCP-1 expression [10,12,29]. RT-qPCR and Western blotting analyses were performed to measure the expression levels of PRDM16 and PGC1a, which regulate the expression of the thermogenic gene UCP-1. We confirmed that Cu B, Cu E, and Cu I increased the mRNA and protein expression levels of PRDM16 and PGC1a in a concentration-dependent manner (Figure 3A,B). These results show that Cu B, Cu E, and Cu I have the potential to produce browned white adipocytes through an increase in UCP-1 expression from upstream regulation.

### 2.4. Effect of Cu B, Cu E, and Cu I on Mitochondrial Biogenesis

The formation of multiple small lipid droplets through lipolysis in adipocytes exhibits brown adipocyte-like properties and leads to increased mitochondrial formation. Furthermore, mitochondrial biosynthesis is the process of generating new mitochondria, and PGC-1a is a key regulator of mitochondrial biosynthesis that can activate the expression of downstream mitochondrial factors, including UCP-1, in various signaling pathways. Brown adipocytes also induce the expression of native UCP-1 in the inner mitochondrial membrane [10,12,27]. To confirm the effect of Cu B, Cu E, and Cu I on mitochondrial biosynthesis, the mitochondrial mass was measured using a confocal microscope after staining with MitoTracker. It was confirmed that Cu B, Cu E, and Cu I increased mitochondrial mass more than 3-fold (Figure 4A). In mature adipocytes, the degree of mitochondrial biosynthesis of Cu B, Cu E, and Cu I was in the order Cu E > Cu B > Cu I. At the same time, it was confirmed by Western blotting that the expression of UCP-1, a mitochondrial inner membrane protein, also increased by more than four times, as confirmed by Western blotting (Figure 4B). These results confirm that Cu B, Cu E, and Cu I upregulated mitochondrial biosynthesis in mature adipocytes and increased the expression of the thermogenic gene UCP-1 at the same time.

### 2.5. Effect of Adipocyte Browning of Cu B, Cu E, and Cu I through PLD2 Inhibition or Knockdown

As a hydrolase, PLD2 breaks down phosphatidylcholine to phosphatidic acid and choline. PLD2 has been reported to be involved in cell migration through cytoskeletal tissue, cell proliferation, death, and differentiation [30,31]. However, the role of PLD2 in the transformation of white adipocytes to brown adipocytes has not yet been clearly investigated. Therefore, to investigate the role of PLD2 in adipocyte browning, CAY10594, a representative PLD2 pharmacological inhibitor, was treated alone or in combination with Cu B, Cu E, and Cu I to investigate adipocyte browning. It was confirmed that the mRNA level of UCP1 increased by 6.6 times compared with that in the control when CAY10594 was used alone. When CAY10594 alone was applied, PRDM16 and PGC1a mRNA levels increased 9.2-fold and 8.9-fold, respectively, compared with those in the control group (Figure 5A). These results confirmed that the expression levels of representative browning markers UCP1, PRDM16, and PGC1a were increased by the PLD2 pharmacological inhibitor. To investigate how the effects of these PLD2 pharmacological inhibitors are modulated by Cu B, Cu E, Cu I, and CAY10594, Cu B, Cu E, and Cu I were treated simultaneously. As a result, it was confirmed that the mRNA expression levels of UCP-1 of Cu B, Cu E, and Cu I were increased by more than seven times compared to the case of treatment with CAY10594 alone. In addition, it was confirmed that the PRDM16 and PGC1a mRNA levels of Cu B, Cu E, and Cu I increased by more than three times and more than four times, respectively, compared to CAY10594-alone treatment (Figure 5A). Next, to investigate the effect of PLD2 pharmacological inhibitors on mitochondrial biosynthesis, MitoTracker staining and confocal microscopy were performed. The intensity of MitoTracker staining increased by 2.8 times compared with that of the control group when CAY10594 alone was used. In addition, it was confirmed that when CAY10594 and Cu B, Cu E, and Cu I were treated simultaneously, Cu B increased 2.6 times, Cu E 3.7 times, and Cu I 2.1 times compared to CAY10594 treatment alone (Figure 5B). We found that the inhibition of PLD activity by CAY10594 affected browning-associated protein expression levels. Since CAY10594 may have a slight non-specific inhibitory effect on PLD2, the expression level of browning-related proteins was confirmed by removing the expression of PLD2 using the Si RNA system. We investigated PLD2 downregulation by using siRNA. The transcription level of PLD2 was significantly downregulated by PLD2-siRNA (si-PLD2); therefore, PLD2-siRNA was used in the subsequent experiments. It was confirmed that the transcription levels of UCP-1, PRDM16, and PGC1a were increased in the si-PLD2 single group, similar to the result of treatment with CAY10594 alone (Figure 5C). In addition, it was confirmed that the transcription levels of UCP-1, PRDM16, and PGC1a increased more than 8-fold when cells were treated with Cu B, Cu E, Cu I, and si-PLD2 (Figure 5D). In mitochondrial biosynthesis, when Cu B, Cu E, Cu I, and si-PLD2 were used together, it was confirmed that the increase was more than five times that of the si-PLD2 group (Figure 5E). From these results, it was confirmed that specific inhibition of PLD2 promotes the browning of white adipocytes, and this effect is significantly increased when Cu B, Cu E, Cu I, and si-PLD2 are treated together.

## 3. Discussion

Over the past several hundred years, bioactive compounds from medicinal plants have emerged as alternatives to conventional obesity treatments. As a potential candidate for the development of anti-obesity drugs, research on the molecular mechanisms of action of compounds derived from medicinal plants is being intensively conducted [4,6]. Recently, natural compounds for the browning of white adipocytes have been intensively investigated [7]. In this study, the molecular biological mechanisms for browning of white adipocytes were investigated using cucurbitacin B, E, and I, which are representative of Cucurbitaceae plants. Recently, it has been demonstrated that pumpkin and watermelon extracts function as PPAR-γ inhibitors to reduce body weight and fat storage in high-fat-diet-induced obese mice. CuB and CuI have been shown to inhibit adipogenesis by inhibiting the differentiation of adipocytes based on STAT3 signaling. Furthermore, CuE has been shown to inhibit visceral obesity, insulin resistance, hyperglycemia, and dyslipidemia in a mouse model through the JAK-STAT5 signaling pathway [23]. However, studies on the efficacy of cucurbitacins B, E, and I on browning of white adipocytes have not yet been clearly demonstrated. In the present study, we successfully generated MDI-induced mature adipocytes. Similar to the previously reported inhibition of adipogenesis by cucurbitacin B, E, and I, lipid accumulation in cucurbitacin B-, E-, and I-treated adipocytes was reduced in a concentration-dependent manner. The inhibitory ability of cucurbitacin B, E, and I on lipid accumulation in white mature adipocytes was in the order of Cu E > Cu B > Cu I. We further confirmed that CuB, CuE, and CuI form small pleiotropic lipid droplets in mature adipocytes, a hallmark characteristic of white adipocyte browning. In addition, the transcription and translation levels of UCP-1, a target marker of beige adipocytes, were increased in CuB, E, and I. PRDM16 and PGC1a are transcription factors that play an important role in inducing differentiation from white adipocytes to beige adipocytes. PRDM16 and PGC1a, which regulate UCP-1 at the upstream signaling site, also demonstrated that CuB, CuE, and CuI increase transcriptional and translational levels. Beige adipocytes have a number of mitochondria around the polytropic lipid droplets, which gives them a characteristic beige color, and CuB, CuE, and CuI have been demonstrated to induce mitochondrial biogenesis.

Studies on adipogenesis and the regulation of PLD, which play important roles in cell proliferation, migration, and differentiation, are still lacking. It was confirmed that PLD1 engages in the differentiation of white adipocytes through mTOR-IRS-1 phosphorylation. Recently, it has been demonstrated that PLD2 has an inverse relationship with UCP1, which is achieved by regulating the number of mitochondria through p62. These results suggest that PLD2 participates in the browning of white adipocytes [30,32]. In this study, we investigated the effect of PLD2 on beige adipocyte properties using a pharmacological inhibitor and knockdown system. We confirmed that the expression of UCP-1, PRDM16, and PGC1a was increased by PLD2 inhibition or knockdown; we also confirmed that mitochondrial biosynthesis was increased. PLD2 inhibition or knockdown on browning of white adipocytes synergistically increased the expression of UCP-1, PRDM16, and PGC1a by CuB, E, and I treatment, and it was also confirmed that mitochondrial biosynthesis was synergistically increased. This demonstrated that the beige adipocyte properties of CuB-, CuE-, and CuI-treated adipocytes were further upregulated through the inhibition of PLD2 or its knockdown.

## 4. Materials and Methods

### 4.1. Reagents and Antibodies

The subsequent reagents were originally purchased from Sigma-Aldrich (Merck KGaA, Darmstadt, Germany): cucurbitacin B, E, and I (Cu B, Cu E, and Cu I; purity > 98% in HPLC), dimethyl sulfoxide (DMSO), Cell Counting Kit-8 (CCK-8), DAPI mounting media, protease inhibitor, and X-treme GENE siRNA transfection reagent. Mouse 3T3-L1 pre-adipocytes were supplied by American Type Culture Collection (ATCC, Manassas, VA, USA). The following reagents for mouse 3T3-L1 pre-adipocytes culture were purchased from GIBCO (Grand Island, NY, USA): phosphate-buffered saline (PBS), Dulbecco’s modified Eagle’s medium/nutrient mixture F-12 (DMEM/F-12), fetal bovine serum (FBS), penicillin/streptomycin, and trypsin-EDTA. The following reagents were obtained from Thermo Fisher Scientific Life Sciences (Waltham, MA, USA): PureLink RNA Mini Kit, high-dose cDNA reverse kit, eight-well chamber slides, SYBR Green qPCR Master Mix, negative control siRNA, PLD2-specific siRNA, MitoTracker Red CMXRos, Pierce ECL Western blotting substrate, and M-PER™ Mammalian Protein Extraction Reagent. The 3T3-L1 Differentiation Kit was purchased from BioVision (Milpitas, CA, USA). The 3T3-L1 differentiation kit was purchased from BioVision (Milpitas, CA, USA), and the Nile red staining kit was acquired from Abcam (Cambridge, MA, USA). Antibody PRDM16 (ab106410) was obtained from Abcam (Cambridge, MA, USA), and UCP1 (sc-293418), PGC1α (sc-518025), and β-actin (sc-47778) were acquired from Santa Cruz.

### 4.2. Pre- and Mature Adipocyte Culture and Treatment

Immobilized murine pre-adipocytes 3T3-L1 were obtained from the American Type Culture Collection (Manassas, VA, USA) and cultured in Dulbecco’s modified Eagle’s medium/nutrient mixture F-12 (DMEM/F-12) with 10% FBS and 1% penicillin/streptomycin at 37 °C in humidified 5% CO_2_ and 95% air. To establish mature adipocytes, pre-adipocytes were cultured until 100% confluence was achieved. To obtain mature adipocytes from pre-adipocytes, MDI differentiation medium (1 μM dexamethasone, 500 μM IBMX, 1.5 μg/mL insulin, and 1 μM rosiglitazone) was added for 3 days. Then, DMEM/F-1 containing 1.5 μg/mL insulin was added for 2 days. Thereafter, the medium was replaced every 2 days, and 9 days after treatment with the total MDI differentiation medium, it was used for each experimental purpose. Cu B, Cu E, and Cu I were dissolved in dimethyl sulfoxide (DMSO) so that the final concentration in the medium was 0.1%, added together during the treatment of MDI differentiation medium, and added every time the medium was replaced for nine days of culture. To evaluate the ability to induce beige adipocyte, Cu B, Cu E, and Cu I (200 nM) were added 2 h before treatment with MDI differentiation medium (Figure 6).

### 4.3. Cell Viability of Pre- and Mature Adipocytes

The viability of Cu B, Cu E, and Cu I in pre-adipocytes and mature adipocytes was determined using the Cell Counting Kit-8 (CCK-8) assay kit (Sigma-Aldrich, St. Louis, MA, USA). Pre-adipocytes were plated in 96-well plates at a density of 3 × 10^3^ cells/well, incubated for 2 days, and then treated with different concentrations of Cu B, Cu E, and Cu I (0, 50, 100, 150, 200, and 300 nM) for 24 h. Mature adipocytes were treated with various concentrations of Cu B, Cu E, and Cu I (0, 50, 100, 150, 200, and 300 nM) under the treatment conditions described in 2.2. Thereafter, 5 µL of Cell Counting Kit-8 (CCK-8) Assay Kit (Sigma-Aldrich-Merck KGaA, Burlington, MA, USA) was added to each well and incubated at 37 °C for 4 h. After incubation, the OD values were measured at 450 nm using a Wallac Victor plate reader (Perkin Elmer Corp., Nerwalk, CT, USA).

### 4.4. qRT-PCR (Quantitative Real-Time PCR)

The quantitative mRNA expression of genes was analyzed using the Bio-Rad Chromo4™ (Applied Biosystems, CA, USA). The PCR primer sequences were as follows (forward and reverse primers, respectively): UCP1, 5′-CCTGCCTCTCTCGGAAACAA-3′, 5′-GTAGCGGGGTTTGATCCCAT-3′; PRDM16, 5′-CAGCACGGTGAAGCCATTC-3′, 5′-GCGTGCATCCGCTTGTG-3′; PGC1α, →5′-ATGTGCAGCCAAGACTCTGTA-3′, 5′-CGCTACACCACTTCAATCCAC-3′; GAPDH, →5′-AGGTCGGTGTGAACGGATTTG-3′, 5′-TGTAGACCATGTAGTTGAGGTCA-3′.

### 4.5. Protein Preparation and Western Blotting Analysis

Total proteins were extracted using a mammalian protein extraction reagent (Thermo Fisher Scientific, Waltham, MA, USA) containing a protease inhibitor (Sigma-Aldrich, USA), and the protein concentrations were measured using a Bradford assay (Bio-Rad Laboratories, Richmond, California). Equal amounts of total protein (30 μg) were loaded onto 7.5% SDS-PAGE gels (Bio-Rad Laboratories, Richmond, California) at 200 V for 40 min, electro transferred to PVDF membranes, blocked in 5% non-fat milk at room temperature for 1 h, and incubated with the diluted primary UCP1 (Santa Cruz: sc-293418), PRDM16 (Abcam: ab106410), PGC1α (Santa Cruz: sc-518025), and β-actin (Santa Cruz: sc-47778) antibodies at 4 °C overnight in turn. After washing three times with PBST, the membranes were probed with horseradish-peroxidase-conjugated secondary antibody at room temperature for 1 h. β-actin was used as the loading control. The protein was detected using Pierce ECL Western blotting substrate (Thermo Fisher Scientific, Waltham, MA, USA), and the protein band density was quantified using ImageQuant TL (version 8.1; Amersham, Amersham, UK).

### 4.6. Evaluation of Lipid Droplets Using Nile-Red Staining

Lipid droplet formation in mature adipocytes was assessed using a Nile red staining kit (Abcam, Cambridge, UK). In summary, 4000 pre-adipocytes were cultured in 8-well chamber slides (Thermo Fisher Scientific Life Sciences) for the evaluation of lipid droplet formation, and Cu B, Cu E, and Cu I treatment conditions were as described in Section 2.2. Mature adipocytes were washed twice with phosphate buffered saline, fixed with 10% formalin, and stained with Nile Red solution at room temperature for 0.5 h. After washing twice with phosphate-buffered saline, the nuclei of mature adipocytes were visualized using Fluoroshield with DAPI (Sigma-Aldrich, USA). After 3–4 drops of Fluoroshield with DAPI were added, the cover slide was covered and incubated at room temperature for 5 min. Images and density of the stained lipid droplets were analyzed using a confocal microscope (LSM 800, Carl Zeiss, Jena, Germany). The degree of Nile red staining of mature adipocytes was quantified based on the fluorescence unit and measured using a flow cytometer (Fit NxT Flow Cytometer; Thermo Fisher Scientific, Inc.).

### 4.7. Immunofluorescence

To detect UCP-1 expression, adipocytes were seeded in 8-well chamber slides. After incubation and Cu B, Cu E, and Cu I treatment using the method described in Section 2.1, the samples were rinsed three times with PBS. After fixation in 4% paraformaldehyde, they were permeabilized for 15 min using 0.1% Triton X diluted in PBS. The adipocytes were then blocked with 5% bovine serum albumin for 1 h at 37 °C, washed with PBS, and then incubated with primary antibody against UCP-1 (1:200) overnight at 4 °C in a humid chamber. The next day, the 8-well chamber slides were carefully rinsed, and the nuclei were counterstained with DAPI loading medium for 5 min. Finally, slides from each of the three fields were randomly selected for microscopy using a Zeiss LSM 800 confocal laser-scanning microscope (Carl Zeiss, Oberkochen, Germany). Relative intensities were acquired using Zeiss ZEN blue edition (Version 3.3.89.0000, Carl Zeiss, Oberkochen, Germany) and ImageJ software (version 1.48) in duplicate.

### 4.8. Mitochondrial Staining and Quantification

Pre-adipocytes were cultured in 8-well chamber slides and processed as described in Section 2.1. Next, adipocytes were incubated with DMEM/F12 containing 50 nM MitoTracker Red CMXRos at 37 °C for 30 min. The 8-well chamber slides were carefully rinsed, and the nuclei were counterstained with DAPI loading medium for 5 min. Finally, slides from each of the three fields were randomly selected for microscopy using a Zeiss LSM 800 confocal laser-scanning microscope (Carl Zeiss, Oberkochen, Germany). Relative intensities were obtained using Zeiss ZEN Blue Edition (version 3.3.89.0000, Carl Zeiss, Oberkochen, Germany) and ImageJ software.

### 4.9. Short Interfering RNA (siRNA) and Transfection

Small interfering RNA (siRNA) was used to confirm the effects of PLD2 on UCP1/PRDM16/PGC1α expression. Double-stranded silencing RNAs for mouse PLD2 (MSS276381) and negative control siRNA were provided by Thermo Fisher Scientific Life Sciences (Waltham, MA, USA) for siRNA experiments. Adipocytes were transfected with mouse siRNAs for PLD2 and control siRNA using the X-treme GENE siRNA transfection reagent, according to the manufacturer’s instructions. The X-treme GENE siRNA transfection reagent, control siRNA, and si-PLD2 were mixed with Opti-MEM I Reduced Serum Medium and then activated for 20 min at room temperature; transfection mixture with control siRNA, and si-PLD2 (100 nM), were then added to the pre-adipocytes. After 24 h of incubation, fresh medium was added and the pre-adipocytes were grown for an additional 48 h. PLD2 levels were analyzed by qRT-PCR to confirm siRNA specificity. siRNA-transfected adipocytes were treated with Cu B, Cu E, or Cu I (200 nM), and qRT-PCR was used to detect UCP1/PRDM16/PGC1α levels.

### 4.10. Statistical Analysis

Data were collected from three independent experiments (n = 3). All values are expressed as the mean ± standard error of measurement (SEM) for three independent exposures to the tested compound. Data are expressed as a percentage of the relevant solvent control for each parameter and were analyzed using Tukey’s post hoc test procedure for multiple comparisons according to one-way ANOVA. Significant effects are represented by *p* ≤ 0.05 (*) and *p* ≤ 0.01 (**).

## 5. Conclusions

In summary, in our study, we explored the roles of CuB, CuE, and CuI on beige adipocyte properties in mature adipocytes. Mechanistic studies indicated that CuB, CuE, and CuI exerted a browning-promoting effect by increasing the expression levels of beige adipocyte marker genes and inducing mitochondrial biogenesis. These effects were further enhanced through the inhibition of PLD2. Our data provide evidence for the role of previously unidentified members of the cucurbitacin family in the browning of white adipocytes, and the possibility that CuB, E, and I may be potential leads for new drugs to prevent obesity by promoting energy expenditure in adipocytes.

## Figures and Tables

**Figure 1 ijms-23-15362-f001:**
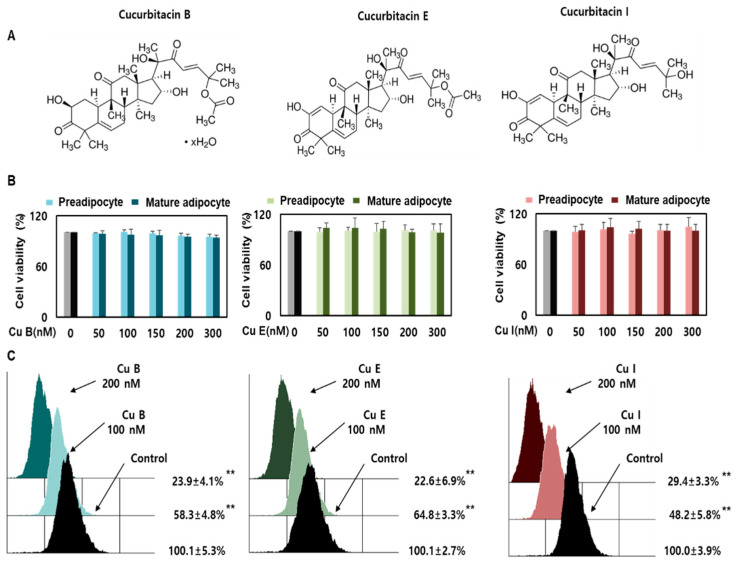
Effects of Cu B, Cu E, and Cu I on cell viability and Nile red-O staining activity. The chemical structures of Cu B, Cu E, and Cu I (**A**). Effects of Cu B, Cu E, and Cu I (50, 100, 150, 200, 300 nM) on the viability of pre-adipocytes and mature adipocytes. Gray is preadipocyte control, black is mature adipocyte control (**B**). Nile-red staining activity of Cu B, Cu E, and Cu I (50, 100, 150, 200, 300 nM) using flow cytometry (**C**). Results are expressed as mean ± SEM (n = 3 per group). ** *p* < 0.01 indicates a statistically significant difference as compared to control group. Cu B, Cucurbitacin B; Cu E, Cucurbitacin E; Cu I, Cucurbitacin I.

**Figure 2 ijms-23-15362-f002:**
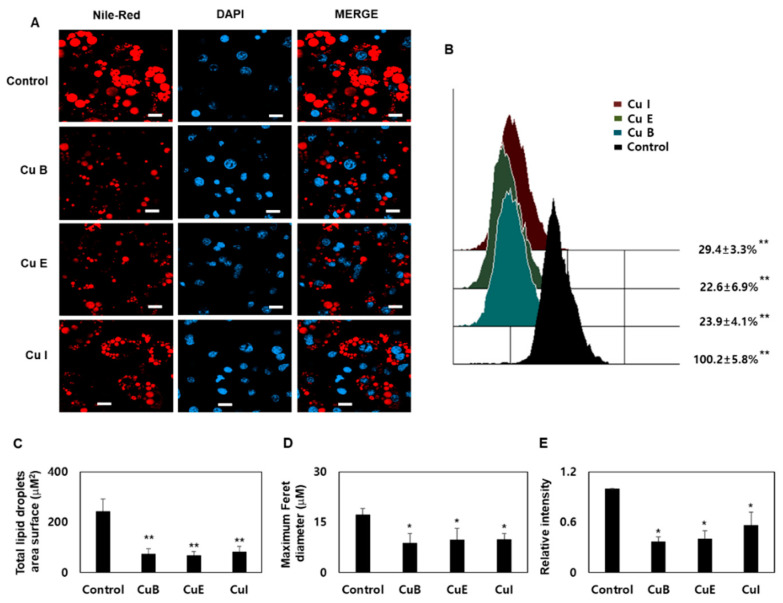
Effect of Cu B, Cu E, and Cu I on lipid droplet formation. Confocal micrographs of Lipid Droplet Morphology of Cu B, Cu E, and Cu I (200 nM), stained with Nile red and DAPI ((**A**), scale bar = 20 μm). Total Nile red staining intensity using flow cytometry after treatment with Cu B, Cu E, and Cu I (200 nM) in mature adipocytes (**B**). Total lipid droplets area surface of Cu B, Cu E, and Cu I (200 nM) using confocal microscopy (**C**). Maximum Feret diameter of Cu B, Cu E, and Cu I (200 nM) using confocal microscopy (**D**). Total Nile red staining intensity of Cu B, Cu E, and Cu I (200 nM) using confocal microscopy (**E**). Results are expressed as mean ± SEM (n = 3 per group). * *p* < 0.05 and ** *p* < 0.01 indicate a statistically significant difference as compared to control group.

**Figure 3 ijms-23-15362-f003:**
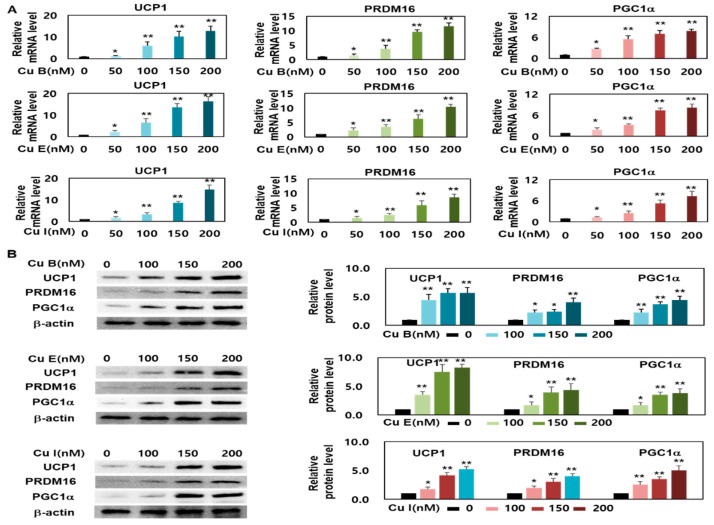
The effects of Cu B, Cu E, and Cu I on mRNA and protein expression levels of UCP1, PRDM16, and PGC1a. (**A**) Transcript expression levels of UCP1, PRDM16, and PGC1a of Cu B, Cu E, and Cu I (50, 100, 150, 200 nM), as measured using RT-qPCR. (**B**) Translation expression levels of aUCP1, PRDM16, and PGC1a, as measured using Western blotting (100, 150, 200 nM). Results are expressed as mean ± SEM (n = 3 per group). * *p* < 0.05 and ** *p* < 0.01 indicate a statistically significant difference as compared to control group.

**Figure 4 ijms-23-15362-f004:**
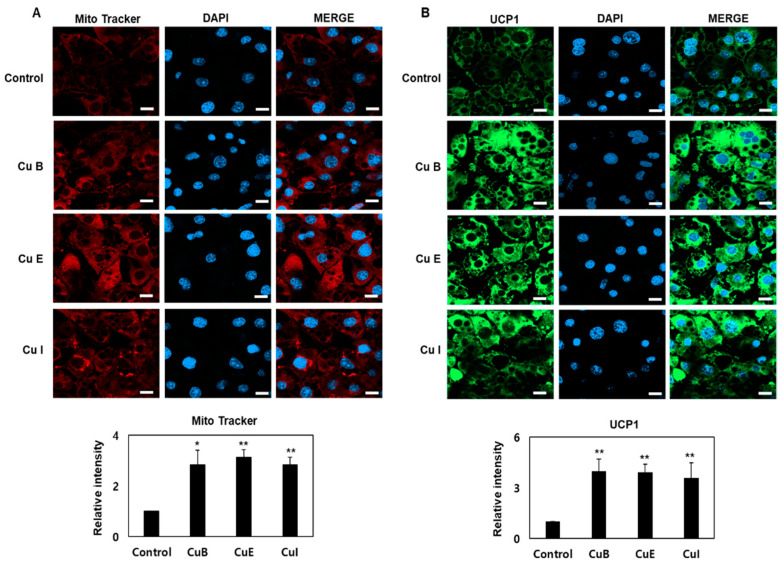
Upregulation of mitochondrial biosynthesis and UCP-1 expression following treatment with Cu B, Cu E, and Cu I. (**A**) Effects of Cu B, Cu E, and Cu I (200 nM) representative confocal micrographs of staining of MitoTracker and DAPI (Top, scale bar = 20 μm), and quantification of staining intensity of MitoTracker (Bottom). (**B**) Effects of Cu B, Cu E, and Cu I (200 nM) representative confocal micrographs of immunofluorescence staining of UCP1 and DAPI (Top, scale bar = 20 μm), and quantification of staining intensity of UCP1 (Bottom). Results are expressed as mean ± SEM (n = 3 per group). * *p* < 0.05 and ** *p* < 0.01 indicate a statistically significant difference as compared to control group.

**Figure 5 ijms-23-15362-f005:**
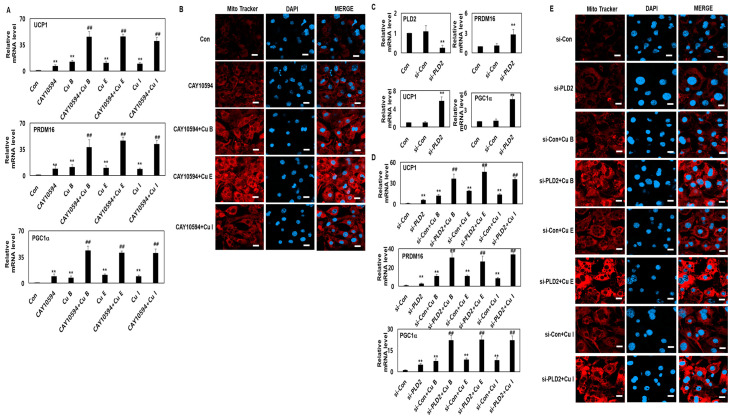
Elevated adipocyte browning levels promoted by Cu B, Cu E, and Cu I via PLD2 down-regulation. (**A**) RT-qPCR results of UCP1, PRDM16, and PGC1a levels in mature adipocytes treated with Cu B, Cu E, and Cu I (200 nM) or CAY10594 (10 mM, scale bar = 20 μm). (**B**) Representative confocal micrographs of MitoTracker and DAPI in mature adipocytes treated with Cu B, Cu E, and Cu I (200 nM) or CAY10594 (10 mM). (**C**) RT-qPCR results of PLD2, UCP1, PRDM16, and PGC1a in mature adipocytes treated with si-PLD2. (**D**) RT-qPCR results of UCP1, PRDM16, and PGC1a in mature adipocyte treated with Cu B, Cu E, and Cu I (200 nM) or si-PLD2. (**E**) Representative confocal micrographs of MitoTracker and DAPI in mature adipocytes treated with Cu B, Cu E, and Cu I (200 nM) or si-PLD2. Results are expressed as mean ± SEM (n = 3 per group). ** *p* < 0.01 and ^##^
*p* < 0.01 indicate a statistically significant difference as compared to CAY10594 group.

**Figure 6 ijms-23-15362-f006:**
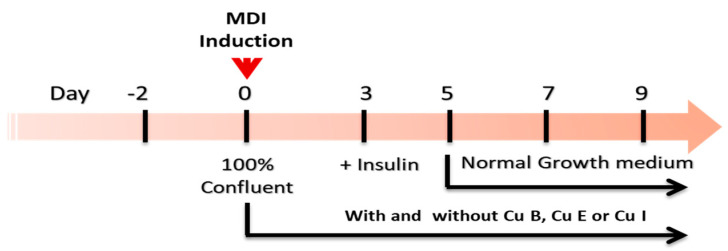
Experimental design of mature adipocyte culture and Cu B, Cu E, and Cu I treatment.

## Data Availability

The data supporting the findings of this study are available from the corresponding author upon reasonable request.

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
