# Peer review of "Cucurbitacin B-, E-, and I-Induced Browning of White Adipocytes Is Promoted by the Inhibition of Phospholipase D2"

_ijms, 2022, doi:10.3390/ijms232315362_

Round 1
Reviewer 1 Report
In this manuscript, authors described the browning effects of cucurbitacins B, E, and I on 3T3-L1 adipocytes and their inhibition of phospholipase D2 is involved. The results are very interesting.
Experimental:
How many hours did the adipocytes treated with cucurbitacins?
Cytotoxic effects were determined after 24-h treatment, but other experiments were not clear.
If the cells were treated with cucurbitacins for 24 and 72 hours, their cytotoxic effects may be stronger.
Table 1 is not necessary. This should be moved into Supporting Information.
Figure 1A. The drawing of chemical structures should be unified. They are slightly different.
Discussion:
Cucurbitacins B and E with a acetyl group were reported stronger cytotoxic effects (less than 1 µM) than cucurbitacin I in several cell lines.
Are there any relations between their cytotoxic effects and browning effects of adipocytes?
Several typing errors are found. Check again please.
Author Response
Thank you very much for allowing us to revise our manuscript entitled, “Cucurbitacin B, E, and I induced browning of white adipocytes is promoted by the inhibition of phospholipase D2”. We appreciate the reviewers for their constructive comments, which were very helpful for improving our paper. The manuscript has been carefully revised according to the reviewers’ comments. The revisions are marked in red in the revised manuscript. The detailed responses (in BOLD type) to the comments are provided below.
In this manuscript, authors described the browning effects of cucurbitacins B, E, and I on 3T3-L1 adipocytes and their inhibition of phospholipase D2 is involved. The results are very interesting.
Experimental:
How many hours did the adipocytes treated with cucurbitacins?
Cytotoxic effects were determined after 24-h treatment, but other experiments were not clear.
If the cells were treated with cucurbitacins for 24 and 72 hours, their cytotoxic effects may be stronger.
Response: In Materials and Methods 4.2 has been modified based on the reviewers' advice.
4.2. Pre- and mature adipocyte culture and treatment
Immobilized murine pre-adipocytes 3T3-L1 were obtained from the American Type Culture Collection (Manassas, VA, USA) and cultured in Dulbecco’s modified Eagle’s medium/ nutrient mixture F-12 (DMEM/F-12) with 10% FBS and 1% penicillin/streptomycin at 37 °C in humidified 5% CO2 and 95% air. To establish the mature adipocytes, To establish mature adipocytes, pre-adipocytes were cultured until 100% confluence was achieved. To obtain mature adipocytes from pre-adipocytes, MDI differentiation medium (1 μM dexamethasone, 500 μM IBMX, 1.5 μg/mL insulin, and 1 μM rosiglitazone) was added for 3 days. Then, DMEM/F-1 containing 1.5 μg/mL insulin was added for 2 days. Thereafter, the medium was replaced every 2 days, and 9 days after treatment with the total MDI differentiation medium, it was used for each experimental purpose. Cu B, Cu E, and Cu I were dissolved in dimethyl sulfoxide (DMSO) so that the final concentration in the medium was 0.1%, added together during the treatment of MDI differentiation medium, and added every time the medium was replaced for nine days of culture. To evaluate the ability to induce beige adipocyte, Cu B, Cu E and Cu I (200 nM) were added 2 hours before treatment with MDI differentiation medium (Fig. 7).
Figure 6. Experimental design of pre- and mature adipocyte culture and PT-AuNS treatment.
Table 1 is not necessary. This should be moved into Supporting Information.
Response: Table 1 was moved to suppliment data according to the reviewer's advice.
Figure 1A. The drawing of chemical structures should be unified. They are slightly different.
Response: Figure 1A has been modified based on the advice of reviewers.
Discussion:
Cucurbitacins B and E with a acetyl group were reported stronger cytotoxic effects (less than 1 µM) than cucurbitacin I in several cell lines.
Are there any relations between their cytotoxic effects and browning effects of adipocytes?
Response: Previous studies have analyzed the adipocyte viability of CuB, CuE, and CuI in pre-adipocytes and mature adipocytes at 0 - 300 nM or less over time. As a result of analyzing the adipocyte viability of CuB, CuE, and CuI, it was confirmed that they did not have any effect on immature adipocytes and mature adipocytes compared to each control group. That is, CuB, CuE or CuI did not show adipocyte toxicity at 300 nM or less. As a result, it was confirmed that CuB, CuE, and CuI upregulated mitochondrial biogenesis and promoted the expression of brown adipogenesis marker genes in mature adipocytes at concentrations of 100 to 200 nM.
Several typing errors are found. Check again please.
Response: The manuscript has been revised according to the advice of the reviewers.
Reviewer 2 Report
In the manuscript entitled “Cucurbitacin B, E, and I induced browning of white adipocytes is promoted by the inhibition of phospholipase D2”, the author investigated the effects of cucurbitacin B, E, and I, and PLD2 inhibition/knockdown on the browning of 3T3-L1 mature white adipocyte. They concluded that both CuB, E, I treatment and PLD2 inhibition/knockdown could induce the browning of 3T3-L1 mature adipocyte, such effect was enhanced when co-treatment. This manuscript suffers from some problems that detract from the overall quality.
1) Previous studies indicated that cucurbitacin B, E, and I could suppress adipocyte differentiation (Cho-Rong Seo, 2014, Food Chem Toxicol; Munazza Murtaza, 2017, Plos One). It would be better to include these studies in the “Introduction”.
2) The underly mechanism by which cucurbitacin B, E, and I regulate the browning of 3T3-L1 mature adipocytes needs to be further explored. Same question for PLD2 inhibition/knockdown induced browning of 3T3-L1 adipocyte. It’s hard to understand why the author combined the browning effects of cucurbitacin B, E, I, and PLD2 inhibition/knockdown in this manuscript, what’s the significance?
3) It would be better to include the reference for the description in line 143 to line 144 “The above results reconfirmed that CuB, CuE, and CuI reduced the accumulation of lipids in mature adipocytes, similar to previously published results”.
4) For Figure 2, the changes in lipid droplets before and after CuB, CuE, and CuI should be provided.
5) Can CuB, E, and I induce lipolysis in 3T3-L1 mature adipocytes?
6) It would be better to explain why the Ucp1 signal was shown in control groups (3T3-L1 white mature adipocyte).
7) It would be better to include more details about the siRNA-mediated knockdown in Figure 5. Which cell type was used (3T3-L1 preadipocyte or mature adipocyte)? siRNA concentration?
Author Response
Thank you very much for allowing us to revise our manuscript entitled, “Cucurbitacin B, E, and I induced browning of white adipocytes is promoted by the inhibition of phospholipase D2”. We appreciate the reviewers for their constructive comments, which were very helpful for improving our paper. The manuscript has been carefully revised according to the reviewers’ comments. The revisions are marked in red in the revised manuscript. The detailed responses (in BOLD type) to the comments are provided below.
In the manuscript entitled “Cucurbitacin B, E, and I induced browning of white adipocytes is promoted by the inhibition of phospholipase D2”, the author investigated the effects of cucurbitacin B, E, and I, and PLD2 inhibition/knockdown on the browning of 3T3-L1 mature white adipocyte. They concluded that both CuB, E, I treatment and PLD2 inhibition/knockdown could induce the browning of 3T3-L1 mature adipocyte, such effect was enhanced when co-treatment. This manuscript suffers from some problems that detract from the overall quality.
1) Previous studies indicated that cucurbitacin B, E, and I could suppress adipocyte differentiation (Cho-Rong Seo, 2014, Food Chem Toxicol; Munazza Murtaza, 2017, Plos One). It would be better to include these studies in the “Introduction”.
Response: Based on the reviewer's comment, we have included it in the introduction.
2) The underly mechanism by which cucurbitacin B, E, and I regulate the browning of 3T3-L1 mature adipocytes needs to be further explored. Same question for PLD2 inhibition/knockdown induced browning of 3T3-L1 adipocyte. It’s hard to understand why the author combined the browning effects of cucurbitacin B, E, I, and PLD2 inhibition/knockdown in this manuscript, what’s the significance?
Response: Studies on adipogenesis and the regulation of PLD, which play important roles in cell proliferation, migration, and differentiation, are still lacking. It was confirmed that PLD1 engages in the differentiation of white adipocytes through mTOR-IRS-1 phosphory-lation. Recently, it has been demonstrated that PLD2 has an inverse relationship with UCP1, which is achieved by regulating the number of mitochondria through p62. These results suggest that PLD2 participates in the browning of white adipocytes [30,32]. Therefore, our research team is currently conducting a research project to elucidate the role of PLD in adipogenesis and white adipocyte browning and will publish a related paper. To summarize the findings so far, deletion of PLD increased adipogenesis, body fat mass, and hepatic steatosis along with upregulation of PPAR-γ and C/EBPa. BAT, WAT, and body fat mass increase were confirmed. It was confirmed that this regulation is made through the Wnt-β-catenin signaling pathway. Based on this result, screening was performed to select PLD target anti-obesity candidates, and it was confirmed that CuB, CuE, and CuI had an indirectly related effect.
- Millar, C.A.; Jess, T.J.; Saqib, K.M.; Wakelam, M.J.; Gould, G.W. 3T3-L1 adipocytes express two isoforms of phospholipase D in distinct subcellular compartments. Biochem Biophys Res Commun 1999, 254, 734-738.
- Kim, H.S.; Park, M.Y.; Yun, N.J.; Go, H.S.; Kim, M.Y.; Seong, J.K.; Lee, M.; Kang, E.S.; Ghim, J.; Ryu, S.H. Targeting PLD2 in adipocytes augments adaptive thermogenesis by improving mitochondrial quality and quantity in mice. J Exp Med 2021, 219, e20211523.
3) It would be better to include the reference for the description in line 143 to line 144 “The above results reconfirmed that CuB, CuE, and CuI reduced the accumulation of lipids in mature adipocytes, similar to previously published results”.
Response: References have been included based on the reviewer's comments.
4) For Figure 2, the changes in lipid droplets before and after CuB, CuE, and CuI should be provided.
Response: Nile Red staining assay was performed using confocal microscopy to measure changes in lipid droplet formation. Lipid droplets act as storehouses for excess neutral lipids, typically triacylglycerols or cholesteryl esters. Abnormal accumulation of lipid droplets was observed in mature adipocytes. Nile red (excitation/emission: 550/640 nm) exhibits intense fluorescence and serves as a sensitive stain for the detection of cytoplasmic lipid droplets. Confocal microscopy is a powerful tool for analyzing lipid droplets via Nile Red staining. The resulting images are presented in Figure 2A, and the changes in lipid droplets before and after CuB, CuE, and CuI were analyzed by comparing control and CuB, CuE, and CuI treatment groups.
5) Can CuB, E, and I induce lipolysis in 3T3-L1 mature adipocytes?
Response: When mature adipocytes were treated with 100 and 200 nM CuB, CuE, and CuI, lipid ac-cumulation decreased in a concentration-dependent manner. A comparison of the lipid accumulation inhibitory levels of CuB, CuE, and CuI confirmed that they had an inhibito-ry effect in the order of CuE>CuB>CuI at 100 nM, and it was confirmed that they had a similarly high inhibitory effect at 200 nM (Figure 1 C). The above results reconfirmed that CuB, CuE, and CuI reduced the accumulation of lipids in mature adipocytes.
6) It would be better to explain why the Ucp1 signal was shown in control groups (3T3-L1 white mature adipocyte).
Response: Published results in several cases have reported the expression of basal UCP1 in 3T3-L1 white mature adipocytes.
7) It would be better to include more details about the siRNA-mediated knockdown in Figure 5. Which cell type was used (3T3-L1 preadipocyte or mature adipocyte)? siRNA concentration?
Response: The manuscript was revised according to the opinions of the reviewers.
4.9. Short interfering RNA (siRNA) and transfection
Small interfering RNA (siRNA) was used to confirm the effects of PLD2 on UCP1/ PRDM16/ PGC1a expression.Double-stranded silencing RNAs for mouse PLD2 (MSS276381) and negative control siRNA were provided by Thermo Fisher Scientific Life Sciences (Waltham, MA, USA) for siRNA experiments. Adipocytes were transfected with mouse siRNAs for PLD2 and control siRNA using X-treme GENE siRNA transfection re-agent, according to the manufacturer’s instructions. The X-treme GENE siRNA transfec-tion reagent, control siRNA, and si-PLD2 were mixed with Opti-MEM I Reduced Serum Medium and then activated for 20 min at room temperature, transfection mixture with control siRNA, and si-PLD2 (100 nM) were then added to the pre-adipocytes. After 24 h of incubation, fresh medium was added and the pre-adipocyte were grown for additional 48 h. PLD2 levels were analyzed by qRT-PCR to confirm siRNA specificity. siR-NA-transfected adipocytes were treated with Cu B, Cu E, or Cu I (200 nM), and qRT-PCR was used to detect UCP1/ PRDM16/ PGC1a levels.
Round 2
Reviewer 2 Report
I have no further comments about the revised manuscript